# Unleash LLMs Potential for Sequential Recommendation by Coordinating Dual Dynamic Index Mechanism

## Abstract

Owing to the unprecedented capability in semantic understanding and logical reasoning, the large language models (LLMs) have shown fantastic potential in developing the next-generation sequential recommender systems (RSs). However, on one hand, existing LLM-based sequential RSs mostly separate the index generation from the sequential recommendation, leading to insufficient integration between the semantic information and the collaborative information. On the other hand, the neglect of the user-related information hinders the LLM-based sequential RSs from exploiting the high-order user-item interaction patterns implicating in user behavior. In this paper, we propose the End-to-End Dual Dynamic ($\mathbf{ED}^2$) recommender, the first LLM-based sequential recommender system which adopts the dual dynamic index mechanism, targeting at resolving the above limitations simultaneously. The dual dynamic index mechanism can not only assembly the index generation and the sequential recommendation into an unified LLM-backbone pipeline, but also make it practical for the LLM-based sequential recommender to take advantage of the user-related information. Specifically, to facilitate the LLMs comprehension ability to the dual dynamic index, we propose a multi-grained token regulator which constructs alignment supervision based on the LLMs semantic knowledge across multiple representation granularities. Moreover, the associated user collection data and a series of novel instruction tuning tasks are specially customized to exploit the user historical behavior in depth and capture the high-order user-item interaction patterns. Extensive experiments on three public datasets demonstrate the superiority of $ED^2$ , achieving an average improvement of 19.62% in Hit-Rate and 21.11% in NDCG metric.

## CCS Concepts

• **Information systems** → **Recommender systems**.

## Keywords

Sequential Recommender Systems, Large Language Models

**ACM Reference Format:**

Anonymous Authors. 2018. Unleash LLMs Potential for Sequential Recommendation by Coordinating Dual Dynamic Index Mechanism. In *Proceedings of Make sure to enter the correct conference title from your rights confirmation emai (Conference acronym 'XX).* ACM, New York, NY, USA, 14 pages. https://doi.org/XXXXXXX.XXXXXXX

**Relevance:** This work is relevant to the focus of *improving the core Web technologies* and the Track of *User modeling, personalization and recommendation.* This work proposes the End-to-End Dual Dynamic recommender system, aiming to unleash the LLMs potential for the field of sequential recommendation.

## 1 Introduction

Sequential recommender systems (RSs) have attracted great research attention from both academia and industry, due to the splendid capability in capturing user interest within the historical behavior [4, 6, 8, 22, 34]. In current literature, sequential recommender systems introduce various deep neural network architectures, including convolutional neural networks (CNNs) [39, 47], recurrent neural networks (RNNs) [24, 38], graph neural networks (GNNs) [31, 45], and transformers [6, 22, 36]. For further improvement on recommendation performance, the pre-trained language models (PLMs) are adopted to capture the semantic information within the attached textual feature [12, 13, 26]. Recently, the emergency of large language models (LLMs) pre-trained on large-scale natural language corpus, such as GPT [33] and LLaMA [41], has revolutionized the sequential recommender systems community [35, 51, 54].

The cornerstone of the LLM-based sequential RSs lies in the integration of two kinds of information, i.e., the collaborative information reflected by the user historical behavior and the semantic information within the attached textual feature. Existing efforts towards developing the LLM-based sequential RSs can be categorized into two main approaches: *(i)* **Content-based** methods [2, 14, 17] verbalize the interaction history into textual sequence (e.g., concatenate the item titles) and then instruct the LLM to directly generate the title of the most possible item. However, such straightforward methods heavily increase the computation expenditure and fail to guarantee legitimate recommendation results [54]. *(ii)* **Index-based** methods [8, 17, 35, 51, 54] incorporate RS and LLM through index mechanism, where each item is associated with an identifier. The compact indices are considered as special tokens and used to extend the LLM vocabulary, aiming to mitigate the computational overhead and guarantee the legitimacy of recommendation results. To be more specific, **pseudo ID based** [8, 17, 54] methods[1] emulate traditional ID paradigm [22, 38, 47] and adopt ID-alike words (e.g., $<item\_1>$) to represent items. Since pseudo ID loses sight of the item semantic information, **semantic ID based** methods [20, 35, 51] introduce vector quantization technique [1, 48] to convert the item textual representation into discrete index (e.g., <4, 1, 3>), which takes the item semantic similarity into account.

Despite remarkable achievements, the current LLM-based sequential RSs still confront with the following limitations [21, 28, 35, 51]. *(i)* The static index mechanism restricts the LLMs in integrating the semantic information and the collaborative information. As illustrated in Figure 1b), existing LLM-based sequential RSs mostly

---

[1] Without ambiguity, the term **ID** and **index (indices)** will be used interchangeably.

adopt the static index mechanism which separates the index generation process from the sequential recommendation process. The static index remains frozen during the recommender optimization and thus disregards the item collaborative similarity [20, 29]. For example, the film *Transformers* (July 3, 2007) and the teaching video *Transformer Detailed Elaboration* (October 28, 2021) are highly similar in terms of the textual contents, yet fractionally overlap within the user interaction records. *(ii)* The ignorance of the user-related information hinders the LLMs from exploiting the high-order user-item interaction patterns. As shown in Figure 1b), most of the leading LLM-based sequential RSs, including FDSA[50], TIGER [35], and LC-Rec[51], conduct the next-item prediction merely depending on the item-related information (i.e., item textual content and interactive item sequence). Without the user-related information, it is impracticable for the LLM-based sequential RSs to capture and utilize the high-order user-item interaction patterns. In traditional sequential RSs [31, 45, 46], the high-order user-item interaction patterns are indispensable and contribute a lot to the recommendation result. For example, the user co-purchase pattern indicates the similar users that share the common interest and the user preference pattern reflects the consistent partiality over a long time-span.

Aiming to address the above limitations simultaneously, we for the first time propose a novel sequential recommender system based on the dual dynamic index mechanism. As illustrated in Figure 1c), the dual dynamic index based sequential RS assembles the indexer and the sequential recommender into an unified streamline. Moreover, the indexer adopts a dual architecture which consists of two homogeneous discrete index generators, taking charge of the index generation of users and items, respectively. Based on the dual dynamic index mechanism, on one hand, the index generation module and the sequential recommender module are conjointly optimized in an end-to-end manner, fusing the semantic information and the collaborative information into a monolithic LLM-backbone. On the other hand, the user-related information (i.e., user textual feature and user-user interaction) is taken into consideration and make it practical for the LLMs to exploit the high-order user-item interaction patterns. However, it is non-trivial to design the dual dynamic index based sequential RS due to the following challenges.

**Un-trained Dynamic User/Item Index Token.** Most of the existing LLM-based RSs merely rely on the sequential recommendation task to improve the LLMs understanding of the static index tokens [35, 42, 51]. Nevertheless, as to the dynamic index based sequential RS, the discrepancy of LLMs comprehension ability to the dynamic index tokens and the natural language tokens is further enlarged, since the dynamic indices are synchronously optimized with the LLM backbone. Therefore, how to boost the LLMs comprehension ability to the dynamic index tokens is challenging.

**Implicit High-order User-Item Interaction Pattern.** The current LLM-based sequential RSs [21, 28, 35, 51] mostly overlook the high-order user-item interaction patterns implicitly contained in the user historical behavior. Typically, the GNN-based sequential RSs excel at modeling the high-order interaction patterns by constructing user behavior graph [31, 43, 45], while the graph structure data is troublesome for the LLMs to utilize. Consequently, how to make it practical for the LLMs to exploit and capture the implicit high-order user-item interaction patterns is also challenging.

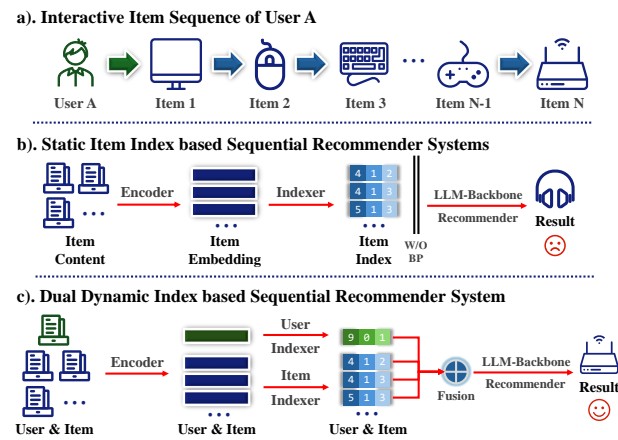

**Figure 1: Overview of a). User interaction sequence, b). Static item index based sequential recommender systems, and c). Dual dynamic index based sequential recommender system. *W/O BP* represents for without back-propagation.**

In this work, we propose the **E**nd-to-**E**nd **D**ual **D**ynamic (**ED**$^2$) recommender system, the first LLM-based sequential recommender system which adopts the dual dynamic index mechanism to accomplish the user/item index generation and the sequential recommendation simultaneously. Horizontally, the ED$^2$ recommender system consists of a dual dynamic index generator and a LLM-backbone recommender. Based on the dual dynamic index mechanism, the ED$^2$ recommender is not only able to seamlessly absorb the semantic information within the textual feature and the collaborative information within the historical interaction, but also can take full advantage of the user-related information. Specifically, to expedite the LLM comprehension ability to the dynamic user/item index tokens, a **m**ulti-**g**rained **t**oken **r**egulator (m-GTR) is proposed to establish different kinds of alignment supervision between the dynamic index tokens and the homologous natural language tokens. Furthermore, we painstakingly construct the associated user collection data based on the user historical behavior and customize a series of novel instruction tuning tasks, targeting at exploiting and utilizing the high-order user-item interaction patterns. Overall, the great potential of LLMs for sequential recommendation is unleashed by the ED$^2$ recommender system. The main contributions of this work are summarized below:

- We for the first time investigate the dual dynamic index based sequential recommender systems and propose the End-to-End Dual Dynamic (ED$^2$) recommender which unleashes the LLMs promising capacity for sequential recommendation task.
- We propose the multi-grained token regulator (m-GTR) to supervise the dynamic index optimization, boosting the LLM comprehension capacity towards the dynamic index tokens.
- We construct the associated user collection from the user behavior and customize several instruction tuning tasks, to sufficiently exploit the high-order user-item interaction patterns.
- Extensive experimental results on three public datasets demonstrate that the ED$^2$ recommender outperforms the SOTA recommender systems, achieving an average improvement of 19.62% in Hit-Rate and 21.11% in NDCG metric.

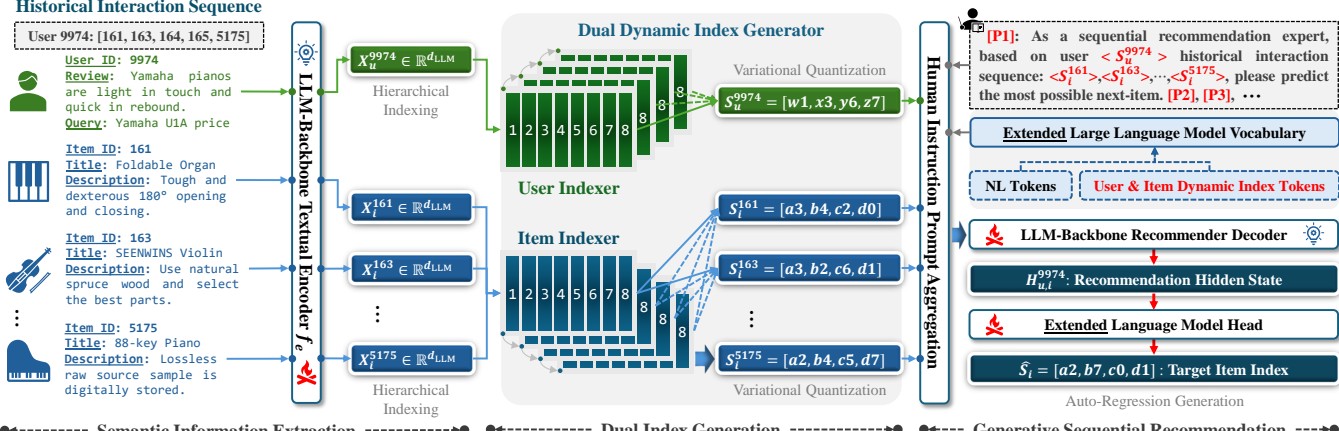

**Figure 2: The overall pipeline of the novel End-to-End Dual Dynamic (ED$^2$) Recommender System. The proposed pipeline includes three stages, i.e., the semantic information extraction, the dual index generation, and the generative sequential recommendation. The semantic information within the user/item attached textual feature is extracted by the LLM-backbone textual encoder, and then the dual dynamic indexer condenses the semantic information into the compact user/item index. Based on the human instruction prompt, the LLM-backbone recommender can directly generate the recommendation result.**

## 2 Methodology

In this section, we introduce the End-to-End Dual Dynamic (ED$^2$) recommender system in detail. As shown in Figure 2, given the user interaction sequence, the LLM-backbone textual encoder first transforms the attached textual features of user and items into textual embeddings. Afterwards, two homogeneous branches within the dual dynamic index generator discretize the embeddings into compact user/item indices. By aggregating the user index and the item indices through the sequential recommendation oriented instruction, the user interaction sequence is reorganized into a heterogeneous sequence composed of dynamic index tokens and natural language tokens. Since the LLM vocabulary has assimilated the dynamic index tokens, the LLM backbone is able to predict the most possible item in an auto-regressive generation manner.

### 2.1 Problem Formulation

We focus on the sequential recommendation task whose target is to predict the most suitable item according to the user historical behavior. Considering a system of $K$ items $\{v_k | k = 1, 2, \cdots, K\}$ and $J$ users $\{u_j | j = 1, 2, \cdots, J\}$, the historical behavior of $u_j$ can be represented by the interactive item sequence $S_j$ as follows,

$$S_j = [v_{k_1}, v_{k_2}, \cdots, v_{k_n}], \quad (1)$$

where $n$ is the sequence length. In addition, we use $T_j^u, T_k^v$ to denote the attached textual feature of user $u_j$ and item $v_k$, respectively. For the item branch, the attached textual content usually includes the item title, the detailed description, and other auxiliaries (e.g., brand and category). Similarly, the user-related textual feature contains the recent comment, the search query, and the holistic user profile. The key notations are summarized in Appendix B for clarity.

### 2.2 End-to-End Dual Dynamic Recommender

The end-to-end dual dynamic (ED$^2$) recommender is composed of a shared LLM backbone and a dual dynamic index generator. The

shared LLM backbone takes charge of understanding the user/item textual feature and reasoning the sequential recommendation result. The dual dynamic index generator is able to quantize the user/item representation provided by the LLM backbone into discrete index. The semantic information is extracted from the textural content with the help of the LLM backbone, and then compressed into the compact index by the dual dynamic index generator, finally integrated with the collaborative information through the sequential recommendation oriented fine-tuning.

*2.2.1 **Semantic Information Extraction**.* To take advantage of the semantic information related to users and items, we initialize the user/item representations on the basis of their attached textual features. For each user $u_j$ and his/her interaction sequence $S_j = [v_{k_1}, v_{k_2}, \cdots, v_{k_n}]$, we look up and organize the corresponding textual features as a collection $\mathcal{T} = \left\{ T_j^u, T_{k_1}^v, T_{k_2}^v, \cdots, T_{k_n}^v \right\}$. Within the LLM-backbone textual encoder $f_e$, the LLM tokenizer first converts the textual content into token indices, and then the token embedding layer projects the token indices into token embeddings. Eventually, the LLM transforms the token embeddings into semantic representations based on the inherent semantic knowledge. The semantic information extraction stage can be formulated as follows,

$$X_T = f_e(T) = \text{LLM}\left(\text{Embed}\left(\text{Tokenizer}\left(T\right)\right)\right) \in \mathbb{R}^d, \quad (2)$$

where $d$ denotes the LLM hidden dimension.

*2.2.2 **Dual Dynamic Index Generation**.* Based on the semantic representations extracted by the LLM-backbone textual encoder, the dual dynamic index generator compresses the semantic information inside to the discrete indices. Due to the discreteness of the dual dynamic index, the downstream LLM-backbone recommender is able to directly generate the index of the recommendation result, adequately stimulating the natural language generation ability of the LLM-backbone. Typically, each user/item is associated with a unique identifier, such as *<user_9974>* or *<item_161>*. A naive

strategy is appending all the unique identifiers into the LLM vocabulary [54], which results in vocabulary exploding linearly with the numbers of users and items. Get inspiration from the sequential quantization technique [12, 48], we adopt a hierarchical architecture in designing the dual dynamic index generator, to represent each user/item with a composition of $M$ index tokens (each token has $N$ possible values). As illustrated in the dual dynamic index generation stage of Figure 2, $<item\_5175>$ can be represented as $\Omega^v_{5175}=<a2, b4, c5, d7>$ with $M$ set to 4 and $N$ set to 8. The expressive space of the hierarchical index mechanism increases exponentially with the index length $M$. The $M$ length hierarchical index with $N$ as the base can theoretically identify $N^M$ distinct objects [12] and the new index tokens are merely $N * M$.

In light of the remarkable performance of the variational auto-encoder architecture in representation compression [10, 15, 48], we design a **h**ierarchical **v**ariational **q**uantization **a**uto-**e**ncoder (HVQAE) to generate the dynamic index. For the $M$ length dynamic index, the HVQAE contains $M$ cascade variational quantizers and quantization codebooks. Specifically, the HVQAE assumes that the dynamic index conforms to the von Mises-Fisher distribution $\text{vMF}(\boldsymbol{\mu}, \kappa)$ which is suitable for the discrete data [37]. The $m$-th variational quantizer $Q_m$ first deduces the statistical characteristics (the mean direction $\boldsymbol{\mu}_m$ and the concentration $\kappa_m$) of the $m$-th dynamic index token, and then the latent token representation $X_{\omega_m}$ is derived according to $\text{vMF}(\boldsymbol{\mu}_m, \kappa_m)$. Afterwards, the dynamic index token $\omega_m$ is sampled based on the categorical distribution over the distance between the latent token representation and the variational quantization codebook $C_m$. As shown in Figure 3, the HVQAE module can be formulated as follows,

$$X_{\omega_m} \sim \text{vMF}(\boldsymbol{\mu}_m, \kappa_m), [\boldsymbol{\mu}_m, \kappa_m] = Q_m(X_m), \quad (3)$$

$$\omega_m \sim \text{Cat}(p_m), p_m = \text{Softmax}\left(\mathcal{D}(X_{\omega_m}, C_m)\right), \quad (4)$$

$$X_{m+1} = X_m - c_m, c_m = \text{One\_Hot}(\omega_m) \cdot C_m, \quad (5)$$

where $m = 1, 2, \cdots, M$ and $X_1 = X_T$, $\text{Cat}(p_m)$ is the categorical distribution over probability $p_m$, $\mathcal{D}$ is the distance metric, and $\text{One\_Hot}(\cdot)$ converts the scalar to a one-hot vector. The dynamic index serves as a hierarchical taxonomy which depicts the corresponding user/item from a coarse-to-fine perspective. The shallow variational quantizer focuses on capturing the general characteristics of user/item and the deep layer pays more attention to the subtle attributes. See Appendix G for the implementation details.

*2.2.3* ***Generative Sequential Recommendation***. By expanding the LLM vocabulary with the collection of the dual index tokens, the $\text{ED}^2$ recommender is able to predict the most possible item in an end-to-end approach. The benefit of adopting the dual dynamic index mechanism for the LLM-based sequential RS development is threefold. *(i)* The dynamic index generator tightly cooperates with the LLM-backbone recommender, to integrate the collaborative information and the semantic information. *(ii)* The index hierarchy captures the semantic information from different granularities and thus is able to indicate the user/item semantic similarity. *(iii)* The index discreteness can reformulate the sequential recommendation task into the language generation task which is familiar to the LLM-backbone pre-trained on similar tasks.

To make the LLM-backbone aware of the sequential recommendation task, we aggregate the dual dynamic indices and the user

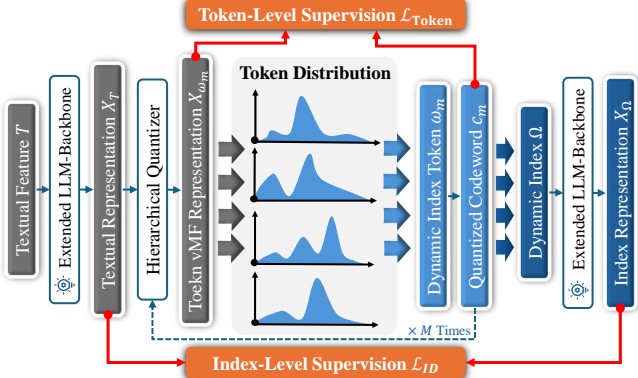

**Figure 3: Overview of the hierarchical variational quantization auto-encoder and the multi-grained token regulator.**

interaction sequence through natural language human instruction. Specifically, the vanilla user ID $u_j$ and item ID $v_k$ in the interaction sequence are replaced with the corresponding dual dynamic indices. Accordingly, the interaction record is reorganized as a heterogeneous sequence composed of the natural language tokens and dual index tokens. An example of the heterogeneous human instruction in our implementation is presented as follows:

> **You are an expert in sequential recommendation. On the basis of the historical interaction sequence:** $\Omega^v_{k_1}, \Omega^v_{k_2}, \cdots, \Omega^v_{k_n}$**, could you please predict the most suitable item for user** $\Omega^u_j$**?**

Denoting the heterogeneous human instruction as $\mathcal{P}$, the LLM backbone first transforms the natural language $\mathcal{P}$ into the hidden representation $X_{\mathcal{P}}$ similar to Formula (2). Then, an extended language model head is appended to project the hidden state $X_{\mathcal{P}}$ into the index token vocabulary, formulated as follows,

$$\hat{\Omega} = \text{LM\_Head}(X_{\mathcal{P}}), \quad (6)$$

where $\hat{\Omega}$ is the index of the recommendation result. If necessary, an inverse look-up operation can identify the vanilla item ID. The sequential recommendation task based on the heterogeneous instruction prompt can be conveniently formulated as a language generation task [40]. The optimization objective is defined as the negative log-likelihood as follows,

$$\mathcal{L}_{\text{LLM}} = -\sum_{i=1}^{B} \log F(\hat{\Omega}_i | \Omega^*_i, \mathcal{P}_i), \quad (7)$$

where $F$ is the combination of the LLM backbone and the extended LM head, $B$ represents the batch-size, $\Omega^*_i$ and $\mathcal{P}_i$ are the ground-truth index and the human instruction of the $i$-th batch, respectively.

## 2.3 Multi-Grained Token Regulator

Since the dual index tokens are considered as special elements of the LLM vocabulary, these tokens serve as the basic notions of a special language to describe the sequential recommendation community. Existing LLM-based sequential recommender systems simply rely on the sequential recommendation oriented fine-tune to

improve the LLM understanding of the index tokens. However, such a straightforward strategy is non-effective for the dynamic index token, since the supervision of sequential recommendation task is unsuitable for the dual dynamic index optimization. Therefore, we propose the multi-grained token regulator (m-GTR) to facilitate the LLM comprehension ability to the dynamic index tokens. As shown in Figure 3, the m-GTR module constructs alignment supervision in both the index level and the token level. To be more specific, given the dynamic index $\Omega$ and the corresponding textual feature $T$, it is noteworthy that $\Omega$ and $T$ describe the same entity from two different perspectives. Hence, the LLM comprehension of the dynamic index $\Omega$ ought to be similar to that of the textual feature $T$. We propose the index level alignment supervision based on the optimization objective as follows,

$$\mathcal{L}_{\text{ID}} = -\frac{1}{B} \sum_{i=1}^{B} \frac{\exp\left(\text{sim}\left(X_\Omega^i, X_T^i\right)\right)}{\sum_{j=1}^{B} \exp\left(\text{sim}\left(X_\Omega^i, X_T^j\right)\right)}, \tag{8}$$

where $B$ is the batch-size, $X_\Omega^i$ and $X_T^i$ are the LLM representation of the dynamic index $\Omega_i$ and the textual feature $T_i$, respectively. Notably, as formulated by Formulas (3)-(5), the dynamic index $\Omega_i$ is composed of $M$ tokens $<\omega_1^i, \omega_2^i, \cdots, \omega_M^i>$. In the $m$-th hierarchy of the variational quantization auto-encoder, the index token $\omega_m$ generated by the quantizer $Q_m$ ought to maintain the user/item similarity. The users/items with similar representations tend to be allocated with similar indices that share a portion of common tokens. Accordingly, we propose the token level alignment supervision based on the optimization objective as follows,

$$\mathcal{L}_{\text{Token}} = -\frac{1}{B}\frac{1}{M} \sum_{i=1}^{B} \sum_{m=1}^{M} \frac{\exp\left(\text{sim}\left(X_m^i, c_m^i\right)\right)}{\sum_{j=1}^{B} \left(w_m^{ij}\right)^{-1} \cdot \exp\left(\text{sim}\left(X_m^i, c_m^j\right)\right)}, \tag{9}$$

$$w_m^{ij} = \text{sim}\left(X_m^i, X_m^j\right), \tag{10}$$

where $X_m^i/X_m^j$ is the input of the $m$-th quantizer and $c_m^i/c_m^j$ is the codeword corresponding to the index token $\omega_m^i/\omega_m^j$. The objective function defined in Formula (9) promotes users/items with similar representations to be assigned similar index tokens, with $w_m^{ij}$ indicating the representation similarity between in-batch samples.

The overall objective function of $ED^2$ is formulated as follows,

$$\mathcal{L}_{\text{All}} = \mathcal{L}_{\text{LLM}} + \beta_1 * \mathcal{L}_{\text{ID}} + \beta_2 * \mathcal{L}_{\text{Token}}, \tag{11}$$

where $\beta_1$ and $\beta_2$ are the weighted hyper-parameters. We present the hyper-parameter analysis towards $\beta_1$ and $\beta_2$ in Appendix H.1.

## 2.4 High-Order Interaction Pattern Exploitation

Most of the current LLM-based sequential recommender systems predict the suitable item merely depend on the item-related information, i.e., the textual feature and the interactive item sequence, while lose sight of the user-related information. Typically, GNN-based sequential recommender systems utilize the user-related information by constructing the user behavior graph, to mine the high-order user-item interaction patterns. Nevertheless, it is complicated for the LLM to deal with the graph structure. Hence, we contrive the high-order user-item interaction pattern exploitation, making it feasible for the LLM to capture these implicit patterns. Specifically, we first construct the associated user collection data on the basis

of the historical behavior. For each item $v_k$, the users who historically interacted with it are recorded as an associated collection $\mathcal{A}_k = \{u_j | v_k \in \mathcal{S}_j\}$, $\mathcal{S}_j$ represents the interaction sequence of $u_j$. Afterwards, a series of instruction tuning tasks are customized for the LLM-backbone recommender to capture the high-order user-item interaction patterns. As a symmetric counterpart to the sequential recommendation task, we design the user prediction task which aims to exploit the user co-purchase pattern. The LLM-backbone recommender is prompted with several users historically interacting with the specific item, and then instructed to generate the index of another user who is fond of the same item. An example of the heterogeneous instruction is presented as follows:

> **You are a professional recommendation system. The item $\Omega_k^v$ is historically purchased by the following users: $\Omega_{j_1}^u, \Omega_{j_2}^u, \cdots, \Omega_{j_n}^u$. Could you please predict another user who may be interested in this item?**

To exploit the user preference pattern, we also design instruction tuning tasks based on the user recent comments, the search queries, and the holistic profile. In detail, the LLM-backbone is prompted with the user historical interaction sequence and instructed to deduce the user comment or the search query, which reflects the implicit user preference to a certain degree. Moreover, the LLM-backbone is instructed to summarize the user profile and the explicit preference based on the interaction history. A detailed summary of the human instructions for the high-order user-item interaction patterns is presented in Appendix D. The designed instruction tuning tasks to exploit the high-order user-item interaction patterns can be formatted as a language generation task as well, with a similar optimization objective to Formula (7).

Overall, to provide a comprehensive understanding to our contributions, we compare the $ED^2$ recommender with three concurrent LLM-based recommender systems [27, 29, 32] in Appendix C. In addition, see Appendix J for the time complexity analysis

## 3 Experiment

In this section, we present and analyse the sequential recommendation performance of the $ED^2$ recommender system on three public datasets. Furthermore, we investigate the effectiveness of the novel $ED^2$ recommender architecture and the superiority of the dual dynamic index mechanism. Finally, we conduct case studies to provide intuitive understanding of the $ED^2$ recommender system. The repository is available at https://anonymous.4open.science/r/ED2-6E01.

## 3.1 Experimental Setup

*3.1.1 Dataset.* We evaluate the proposed $ED^2$ recommender system and all the baseline models on public benchmarks from the Amazon Product Review dataset [11], containing user review data from May 1996 to October 2018. Particularly, we extract three categories for the sequential recommendation task, i.e., *"Musical Instruments"*, *"Video Games"*, and *"Arts, Crafts and Sewing"*. Following standard procedure [35, 51], inactive users/items with less then 5 interactions are filtered out and the user interactive sequence is created according to the chronological order. The dataset statistics and detailed pre-processing are presented in Appendix E.

**Table 1: Performance comparison of ED$^2$ recommender and baselines on three public datasets. The best and the runner-up performance are indicated in bold and underlined font, respectively. For evaluation stability, the presented performance of ED$^2$ recommender is the average result over several distinct instructions. The *Improvement* is defined as (Best-Second)/Second and the superscript * indicates the improvement is statistically significant with the p-value less than 0.01.**

| Dataset | Instruments | | | | | Games | | | | | Arts | | | | |
|---|---|---|---|---|---|---|---|---|---|---|---|---|---|---|---|
| Metric | HR@1 | HR@5 | HR@10 | NDCG@5 | NDCG@10 | HR@1 | HR@5 | HR@10 | NDCG@5 | NDCG@10 | HR@1 | HR@5 | HR@10 | NDCG@5 | NDCG@10 |
| FMLP-Rec | 0.0480 | 0.0786 | 0.0988 | 0.0638 | 0.0704 | 0.0152 | 0.0571 | 0.0930 | 0.0361 | 0.0476 | 0.0310 | 0.0757 | 0.1046 | 0.0541 | 0.0634 |
| Caser | 0.0149 | 0.0543 | 0.0710 | 0.0355 | 0.0409 | 0.0085 | 0.0367 | 0.0617 | 0.0227 | 0.0307 | 0.0138 | 0.0379 | 0.0541 | 0.0262 | 0.0313 |
| HGN | 0.0523 | 0.0813 | 0.1048 | 0.0668 | 0.0744 | 0.0154 | 0.0517 | 0.0856 | 0.0333 | 0.0442 | 0.0300 | 0.0622 | 0.0875 | 0.0462 | 0.0544 |
| GRU4Rec | 0.0571 | 0.0821 | 0.1031 | 0.0698 | 0.0765 | 0.0176 | 0.0586 | 0.0964 | 0.0381 | 0.0502 | 0.0421 | 0.0749 | 0.0964 | 0.0590 | 0.0659 |
| SR-GNN | 0.0538 | 0.0805 | 0.1022 | 0.0673 | 0.0743 | 0.0154 | 0.0554 | 0.0900 | 0.0355 | 0.0466 | 0.0419 | 0.0720 | 0.0926 | 0.0574 | 0.0638 |
| MA-GNN | 0.0296 | 0.0545 | 0.0735 | 0.0497 | 0.0569 | 0.0121 | 0.0371 | 0.0610 | 0.029 | 0.0384 | 0.0188 | 0.0403 | 0.0559 | 0.0353 | 0.0412 |
| BERT4Rec | 0.0435 | 0.0671 | 0.0822 | 0.0560 | 0.0608 | 0.0136 | 0.0482 | 0.0763 | 0.0311 | 0.0401 | 0.0337 | 0.0559 | 0.0713 | 0.0451 | 0.0500 |
| SASRec | 0.0503 | 0.0751 | 0.0947 | 0.0627 | 0.0690 | 0.0145 | 0.0581 | 0.0940 | 0.0365 | 0.0481 | 0.0225 | 0.0757 | 0.1016 | 0.0508 | 0.0592 |
| FDSA | 0.0520 | 0.0834 | 0.1046 | 0.0681 | 0.0750 | 0.0161 | 0.0644 | 0.1041 | 0.0404 | 0.0531 | 0.0451 | 0.0734 | 0.0933 | 0.0595 | 0.0660 |
| S3-Rec | 0.0367 | 0.0863 | 0.1136 | 0.0626 | 0.0714 | 0.0119 | 0.0606 | 0.1002 | 0.0364 | 0.0491 | 0.0245 | 0.0767 | 0.1051 | 0.0521 | 0.0612 |
| P5-CID | 0.0587 | 0.0827 | 0.1016 | 0.0708 | 0.0768 | 0.0177 | 0.0506 | 0.0803 | 0.0342 | 0.0437 | 0.0485 | 0.0724 | 0.0902 | 0.0607 | 0.0664 |
| TIGER | 0.0608 | 0.0863 | 0.1064 | 0.0738 | 0.0803 | 0.0188 | 0.0599 | 0.0939 | 0.0392 | 0.0501 | 0.0465 | 0.0788 | 0.1012 | 0.0631 | 0.0703 |
| CLLM4Rec | 0.0336 | 0.0666 | 0.0845 | 0.0516 | 0.0574 | 0.0142 | 0.0421 | 0.0650 | 0.0282 | 0.0355 | 0.0369 | 0.0724 | 0.0933 | 0.0555 | 0.0621 |
| LC-Rec LoRA | 0.0576 | 0.0817 | 0.1009 | 0.0698 | 0.0760 | 0.0165 | 0.0520 | 0.0835 | 0.0342 | 0.0443 | 0.0367 | 0.0637 | 0.0837 | 0.0504 | 0.0569 |
| ED$^2$ (Ours) | **0.0714***| **0.1028***| **0.1281***| **0.0872***| **0.0947***| **0.0254***| **0.0704***| **0.1083***| **0.0480***| **0.0597***| **0.0639***| **0.1002***| **0.1260***| **0.0823***| **0.0906*** |
| Improvement | 17.43% | 19.12% | 12.76% | 18.16% | 17.93% | 35.11% | 9.32% | 4.03% | 18.81% | 12.43% | 31.75% | 27.16% | 19.89% | 30.43% | 28.88% |

*3.1.2 Baseline.* To comprehensively demonstrate the multifaceted superiority of ED$^2$ recommender, the evaluation includes the following baselines based on various methodologies. **MLP-Based**: FMLP-Rec [53], **CNN-Based**: Caser [47], **RNN-Based**: HGN [30] and GRU4Rec [19], **GNN-Based**: SR-GNN [45] and MA-GNN [31], **Transformer-Based**: BERT4Rec [36], SASRec [22], FDSA [50] and S$^3$-Rec [52], and **PLM/LLM-Based**: P5-CID [8], TIGER [35], CLLM4Rec [54] and LC-Rec [51]. A detailed introduction to the above baselines is presented in Appendix F.

*3.1.3 Evaluation Strategy.* We adopt the Top-K Hit-Rate (HR@K) with K = 1, 5, 10 and the Normalized Discounted Cumulative Gain (NDCG@K) with K = 5, 10 to evaluate the sequential recommendation performance. Following standard setting, the leave-one-out strategy is adopted [35, 51]. Specifically, the most recent item serves as the evaluation data, the second most recent item servers as the validation data, and the remaining interactive items form the training data. The implementation detail and the hardware environment of the ED$^2$ recommender are presented in Appendix G.

## 3.2 Main Result

The evaluation results of ED$^2$ recommender system and the SOTA baselines on three public datasets are presented in Table 1 and the following conclusions can be derived.

**First, integrating semantic information and collaborative information is effective to improve sequential recommendation performance.** For the evaluated models, the sequential recommender systems incorporating the semantic information with the collaborative information (i.e., FDSA, S3-Rec, P5-CID, TIGER, CLLM4Rec, LC-Rec, and ED$^2$ ) achieve an overall superiority, compared with the traditional sequential recommender systems which merely rely on the collaborative information (i.e., FMLP-Rec, Caser, HGN, GRU4Rec, SR-GNN, MA-GNN, BERT4Rec, and SASRec). Compared with the best traditional sequential recommender system GRU4Rec, ED$^2$ ameliorates the recommendation performance up to 29.67% in Hit-Rate and 28.44% in NDCG metric. **Second, dual**

**dynamic index mechanism unleashes LLM potential for sequential recommendation.** In general, the proposed ED$^2$ recommender system constantly outperforms the comparison baselines across the three datasets. Compared with the SOTA LLM-based sequential recommender systems that adopt the static index mechanism (i.e., P5-CID, TIGER, CLLM4Rec, and LC-Rec), ED$^2$ recommender achieves an average improvement of 19.56% in Hit-Rate and 21.11% in NDCG metric. The superiority of ED$^2$ demonstrates that the index generator ought to cooperate with the sequential recommender, unleashing the LLM potential for sequential recommendation. **Third, specific instruction tuning for high-order user-item interaction patterns is crucial.** As to the baselines fusing the semantic information and the collaborative information, only CLLM4Rec underscores the user-related information. However, the CLLM4Rec recommender fails to achieve a significant performance improvement. The degradation can be attributed to the absence of instruction tuning task which specifically exploits the high-order user-item interaction patterns. The ED$^2$ recommender outperforms CLLM4Rec by 72.80% in Hit-Rate and 66.97% in NDCG metric, which reveals the importance of specially customized tuning tasks in utilizing the high-order user-item interaction patterns.

## 3.3 Ablation Study

*3.3.1 Architecture.* To demonstrate the effectiveness of the innovative architecture, including the dual dynamic index mechanism, the multi-grained token regulator, and the high-order user-item interaction pattern exploitation, we conduct ablation study to investigate the contribution of each module. Accordingly, ED$^2$ refers to the unabridged model, *w/o user* removes the user related modules including the user dynamic index generator and the high-order user-item interaction pattern exploitation tasks, *w/o dynamic* freezes the user/item indices during the sequential recommendation optimization, *w/o m-GTR* removes the multi-grained token regulator, *w/o exploit* removes the specific tuning tasks for the high-order user-item interaction patterns, and *w/o u&m* removes both the user related modules and the multi-grained token regulator.

**Table 2: Ablation study towards the architecture innovation. *P-Baseline* records the prime baseline performance which is the best among all the evaluated baselines. *Avg.D* is the average performance degradation over the evaluated metrics.**

| Metric | HR@1 | HR@5 | HR@10 | NDCG@5 | NDCG@10 | Avg.D |
|---|---|---|---|---|---|---|
| $ED^2$ | **0.0714** | **0.1028** | **0.1281** | **0.0872** | **0.0947** | N/A |
| w/o user | 0.0701 | 0.1010 | 0.1230 | 0.0860 | 0.0929 | 2.17% |
| w/o dynamic | 0.0621 | 0.0828 | 0.1005 | 0.0727 | 0.0784 | 17.57% |
| w/o m-GTR | 0.0685 | 0.0986 | 0.1224 | 0.0835 | 0.0911 | 4.13% |
| w/o exploit | 0.0696 | 0.1007 | 0.1236 | 0.0852 | 0.0926 | 2.52% |
| w/o u&m | 0.0610 | 0.0919 | 0.1159 | 0.0769 | 0.0845 | 11.46% |
| P-Baseline | 0.0608 | 0.0863 | 0.1136 | 0.0738 | 0.0803 | 14.56% |

**Table 3: Ablation study towards the index mechanism.**

| Metric | HR@1 | HR@5 | HR@10 | NDCG@5 | NDCG@10 | Avg.D |
|---|---|---|---|---|---|---|
| $ED^2$ | **0.0714** | **0.1028** | **0.1281** | **0.0872** | **0.0947** | N/A |
| 3-$ED^2$ | 0.0686 | 0.1004 | 0.1222 | 0.0844 | 0.0914 | 3.51% |
| 5-$ED^2$ | 0.0697 | 0.1007 | 0.1233 | 0.0853 | 0.0926 | 2.51% |
| Static | 0.0621 | 0.0828 | 0.1005 | 0.0727 | 0.0784 | 17.57% |
| D-LSH | 0.0677 | 0.0976 | 0.1228 | 0.0829 | 0.0909 | 4.66% |
| S-LSH | 0.0660 | 0.0918 | 0.1131 | 0.0790 | 0.0859 | 9.73% |
| P-Baseline | 0.0608 | 0.0863 | 0.1136 | 0.0738 | 0.0803 | 14.56% |

According to the result in Table 2, we derive the following three conclusions. *(i)* **m-GTR module facilitates the LLM comprehension on the dynamic index tokens.** The 4.13% performance reduction of the *w/o m-GTR* variant demonstrates the effectiveness of m-GTR in facilitating the LLM understanding to the index tokens. The same conclusion can be drawn from the performance gap between the *w/o user* variant and the *w/o u&m* variant as well. *(ii)* **Introduction of user-related information aggravates the limitation of static index mechanism.** The *w/o dynamic* variant that replaces the dynamic index mechanism with static counterpart is the most inferior among all the variants, even worse than the best baseline performance. Compared with the single-branch variants (i.e., *w/o user* and *w/o u&m*), the user branch makes it further difficult for the static index mechanism to fuse the semantic information with the collaborative information, leading to the severe performance degradation up to 17.57%. *(iii)* **Naive utilization of user-related information is profitless to sequential recommendation task.** Moreover, comparing the *w/o exploit* variant with the *w/o user* variant, one may notice that simply introducing the user-related information without corresponding instruction tuning task fails to improve the recommendation performance, yet increases the difficulty of sequential recommender optimization.

*3.3.2 Indexing Mechanism.* Furthermore, we investigate the impact of different index mechanisms. First, we implement an $ED^2$ variant that adopts static index mechanism, denoted as *Static*. Then, we introduce dynamic locality sensitive hashing (LSH) index and static LSH [18] index (denoted as *D-LSH* and *S-LSH*) as the comparison variants. Moreover, since the $ED^2$ recommeder in main experiment adopts dynamic index consisting of 4 tokens, we further implement two $ED^2$ variants with the dynamic index length set to 3 and 5 respectively (denoted as 3-$ED^2$ and 5-$ED^2$). The evaluation result on Instruments dataset is presented in Table 3. See Appendix I for the details of the different indexing mechanisms.

We can summarize the following conclusions from the presented result. *(i)* **Representing each item/user with 4 index tokens**

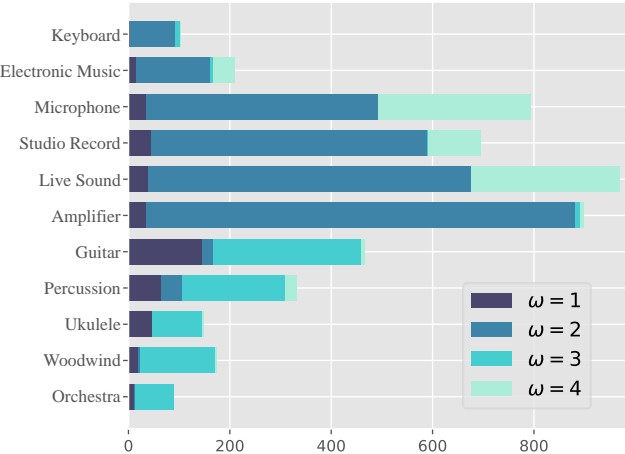

**Figure 4: Category distribution of the 1-st index token $\omega$.**

**is appropriate in such a data magnitude (10K $\sim$ 45K).** 3-$ED^2$ (3.51%↓) with less semantic index length may suffer from inadequate expressiveness and 5-$ED^2$ (2.51%↓) with larger representation space will increase the difficulty of target item generation. *(ii)* **Superiority of dynamic index mechanism is loosely coupled with the indexing method.** Similar to the predominance of $ED^2$ over the *Static* variant, the improvement of *D-LSH* over the static counterpart *S-LSH* further verifies the effectiveness of the dynamic index mechanism, which is feasible to the LSH index as well.

## 3.4 Case Study

To provide more intuitions about the proposed $ED^2$ recommender, we conduct case studies towards *(i)* the category distribution of dynamic index token, *(ii)* the integration ability of the semantic information and the collaborative information.

*3.4.1 Category Distribution of Dynamic Index.* For exposition, we implement a variant $ED^2$ recommender whose index base is set as [4, 32, 256, 256]. Figure 4 illustrates the categorical distribution of the first dynamic index token $\omega$. The $y$-axis is the ground-truth category in the Instruments dataset and the $x$-axis represents the item number. It is noteworthy that the first index token $\omega$ is able to capture category feature of items. Concretely, $\omega = 1, 4$ mainly represent the non-electronic instruments and $\omega = 2, 3$ mainly represent the electronic instrument accessories. More specifically, majority of *Woodwind* and *Orchestra* are represented by $\omega = 3$, most items of *Keyboard*, *Electronic Music*, and *Amplifier* are included in $\omega = 2$.

*3.4.2 Integration Ability.* In Figure 5, we present three cases of sequential recommendation hits, sequential recommendation misses, and user preference summarization, to investigate the integration ability of $ED^2$. In case 1, $ED^2$ succeeds in predicting the most possible item (i.e., the ground-truth item <$a1,b155,c222,d138$>) and the item title generated by $ED^2$ is accord with the ground-truth title. Moreover, we present other recommendation results with lower scores provided by beam search. One can note that these results are tightly relevant to the ground-truth item and $ED^2$ can generate their titles correctly. In case 2 where $ED^2$ fails to recommend the ground-truth item, $ED^2$ can still provide correct comprehension on item semantic information. However, $ED^2$ makes a mistake in

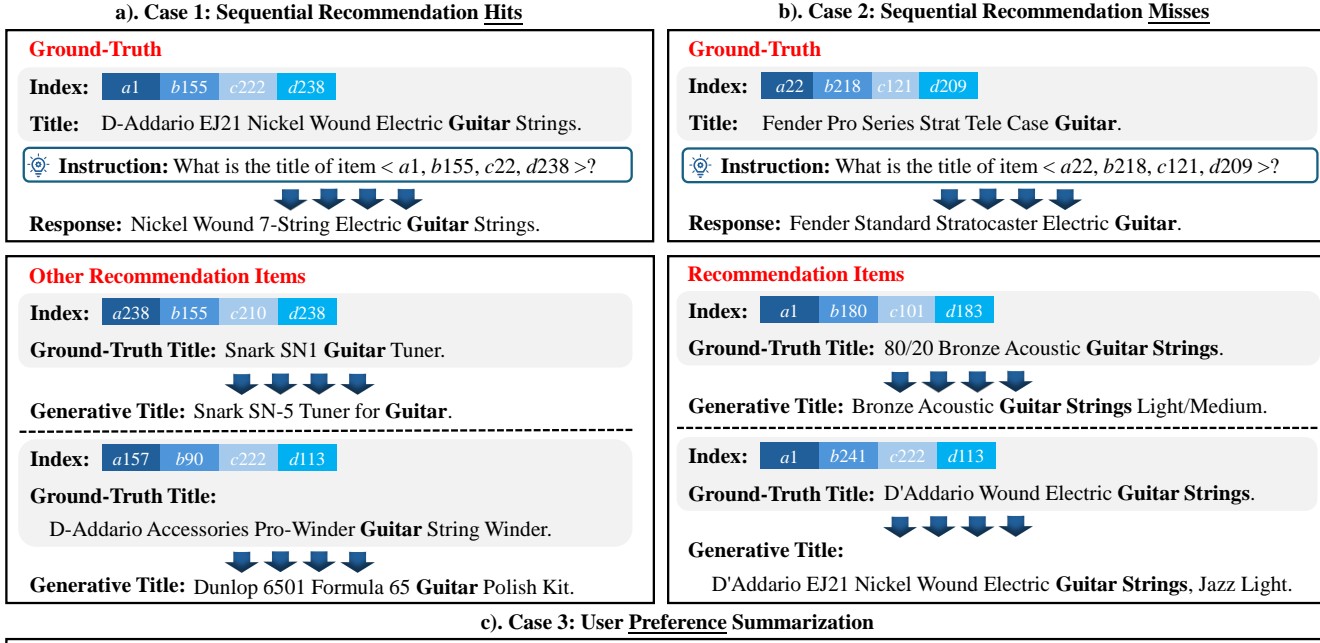

**Figure 5: The case study where ED$^2$ a). hits the sequential recommendation, b). misses the sequential recommendation, and c). summarizes the user preference. The sequential recommendation results are generated by beam search.**

deducing the user interest, dispatching from the instrument *Guitar* to the correlated accessory *Guitar String*. We present the capacity of user preference summary in case 3. Comparing the response with the user historical interaction, we can note that ED$^2$ accurately describes the user's preference in guitar accessories.

## 4 Related Work

**Sequential Recommendation.** Artificial neural network based deep sequential recommender systems have dominated the current leaderboard [7]. GRU4Rec [19] for the first time incorporates GRU module [3] with sequential recommendation. NARM [25] further enhances the long-term user preference memory through attention mechanism, and AttRec [49] introduces metric learning to capture user-item affinity additionally. Inspired by the great success of pretrained language models and masked token prediction, Bert4Rec [36] first utilizes deep transformer model with masking pre-train strategy on sequential recommendation. On the other hand, generative recommendation models based on index learning technique have attracted significant research attention, due to the efficient inference. DSI [40] for the first time proposes an end-to-end generative model for document retrieval, NCI [44] further supplements the indexing mechanism of DSI by proposing a specific prefix decoder.

**LLM-based Recommender System.** As the LLMs demonstrating comprehensive capacity on language modeling tasks, P5 [8]

for the first time attempts to fine-tune LLM for a sequential recommendation oriented model, M6-Rec [5] replaces the pseudo index in P5 with the corresponding linguistic description, and TALL-Rec [2] combines both the pseudo index and textual description. CLLM4Rec [54] extends the LLM vocabulary with pseudo item index and proposes a mixed prompting strategy to adapt the extended LLM. Furthermore, TIGER [35] for the first time adopts hierarchical vector quantization technique to generate semantic item index and LC-Rec [51] substitutes the T5X backbone with LLaMA [41] for superior recommendation performance.

## 5 Conclusion

We for the first time investigate the LLM-based sequential recommender systems that adopts dual dynamic index mechanism and propose the End-to-End Dual Dynamic (**ED$^2$**) recommender system. To unleash the LLM capacity for sequential recommendation, ED$^2$ conjointly optimizes the index generator and the sequential recommender. We further design a multi-grained token regulator to facilitate the LLM comprehension ability to the dual dynamic index tokens. Moreover, we construct associated user collection and customize a series of instruction tuning tasks, to exploit and utilize the high-order user-item interaction patterns. Extensive experiments on three public datasets demonstrate the superiority of the proposed ED$^2$ recommender over the SOTA baselines.

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

# A Background

In this section, we briefly introduce the sequential recommendation task and the large language model.

**Sequential Recommendation.** By analyzing the user historical interactions, sequential recommendation aims to identify user preference and predict the suitable item that would be engaged with [6, 22, 24, 36, 38, 39, 45–47]. Given a chronologically organized sequence of interacted items $S = \{v_1, v_2, \cdots, v_n\}$, the objective function of sequential recommender system $f$ is to maximize the corresponding likelihood defined as follows,

$$\log p(v_1, v_2, \cdots, v_n; f) = \sum_{i=1}^{n-1} \log f(v_{i+1}|v_1, v_2, \cdots, v_i). \quad (12)$$

Finally, the trained sequential recommender system $f^*$ is optimized by maximizing the likelihood function over all the $J$ training interaction sequences, that is,

$$\begin{aligned}
f^* &= \arg\max_f \sum_{j=1}^{J} \log p(S_j; f) \\
&= \arg\max_f \sum_{j=1}^{J} \sum_{i=1}^{|S_j|-1} \log f(v_{i+1}|v_1, v_2, \cdots, v_i).
\end{aligned} \quad (13)$$

**Large Language Model.** Transformer-based models with billions learnable parameters trained on large scale corpora [9, 33, 41], i.e., large language models (LLMs), have presented astonishing capabilities in natural language understanding and logical reasoning based on learned knowledge. The mainstream LLMs mostly belong to the decoder-only architecture with superior generative ability [33], which consists of a token embedding layer, a single decoder module, and a correlated tokenizer. Given a natural language sentence $S = \{w_1, w_2, \cdots\}$, the tokenizer first converts the sentence into sequence of token index $t_i$ whose corresponding word $w_i$ is included in the LLM vocabulary, formulated as follows,

$$\begin{aligned}
T = \{t_1, t_2, \cdots\} &= \text{Tokenizer}(S) \\
&= \{\text{Tokenizer}(w_1), \text{Tokenizer}(w_2), \cdots\}.
\end{aligned} \quad (14)$$

Then, the token index sequence $T$ is fed into the token embedding layer and projected into continuous latent space, by the table look-up operation as follows,

$$H = \text{Embed\_Token}(T) = \text{One\_Hot}(T) \cdot E, \quad (15)$$

$$\text{One\_Hot}(T) \in \{0, 1, \cdots, |\mathcal{V}| - 1\}^{L \times |\mathcal{V}|}, E \in R^{|\mathcal{V}| \times d},$$

where $E$ is the token embedding table, $L$ is the token sequence length, $\mathcal{V}$ is the LLM vocabulary, and $d$ is the LLM hidden dimension. Eventually, the continuous embedding $H$ forwards through the LLM decoder module. In addition, for the LLM aiming at generation task, a language model head is appended to transform the final representation $H'$ back into the token index space, as follows,

$$H' = \text{Decoder}(H), T' = \text{LM\_Head}(H'). \quad (16)$$

Afterwards, the tokenizer is able to translate the newly generated token index sequence $T'$ into natural language sentence $S'$ by an inverse table look-up operation as opposed to Formula (14).

# B Basic Notation

The basic notations and descriptions are summarized in Table 4.

**Table 4: Basic notations and descriptions in the manuscript.**

| Notation | Description |
|---|---|
| $v_k, K$ | $k$-th item, number of items |
| $u_j, J$ | $j$-th user, number of users |
| $S_j$ | Interaction sequence of the $j$-th user |
| $T_k^v, T_j^u$ | Textual feature of the $k$-th item, $j$-th user |
| $f_e$ | LLM-backbone textual encoder |
| $d$ | LLM hidden dimension |
| $\Omega$ | Dynamic index |
| $M$ | Index length, layers of HVQAE module |
| $N$ | Index base |
| $\mu, \kappa$ | vMF mean direction and concentration |
| $\omega_m$ | $m$-th dynamic index token |
| $Q_m$ | $m$-th variational quantizer |
| $C, c$ | Learnable codebook, codeword |
| $\mathcal{P}$ | Human instruction prompt |
| $X_{\mathcal{P}}$ | LLM Representation of human instruction |
| $F$ | LLM backbone with a language model head |
| $B$ | Batch-size |
| $w^{ij}$ | In-batch representation similarity |
| $\beta_1, \beta_2$ | Weight of index/token level alignment |
| $\mathcal{A}_k$ | Associated user collection of item $v_k$ |

# C Discussion

In this section, we compare $ED^2$ with three concurrent LLM-based RSs, i.e., ETEGRec [27], STORE [29], and TokenRec [32].

## C.1 $ED^2$ vs. ETEGRec

The concurrent work ETEGRec [27] introduces iterative training strategy to optimize the item index generator (i.e., indexer) and the generative sequential recommender. In addition, ETEGRec designs alignment between the target item and the historical items, to regulate the iterative optimization. Compared with $ED^2$, the limitations of ETEGRec is twofold. On one hand, the iterative optimization leads to insufficient integration of the semantic information and the collaborative information. On the other hand, ETEGRec still neglects the impact of the user-related information.

## C.2 $ED^2$ vs. STORE

Similar to the $ED^2$ recommender, the concurrent work STORE [29] also constructs a unified pipeline that streamlines the item index generation and the sequential recommendation using a single LLM. However, STORE loses sight of the user-related information as well, failing to capture the high-order user-item interaction patterns within the historical behavior.

## C.3 $ED^2$ vs. TokenRec

Another concurrent work TokenRec [32] emphasizes the high-order user-item knowledge and proposes a twin-tower tokenizer which is similar to the dual dynamic indexer in $ED^2$ recommender system. Nevertheless, TokenRec fails to break through the constraint of the static index mechanism, freezing the user/item indices during the sequential recommender optimization. To sum up, we present an overall comparison against several leading LLM-based recommender systems in Table 5.

**Table 5: Comparison of the ED$^2$ recommender with several leading LLM-based generative recommender systems.**

| Method | Backbone | Index Mechanism | Interaction Pattern | Token Alignment | Collision Handle |
|--------|----------|-----------------|---------------------|-----------------|------------------|
| CLLM4Rec [54] | GPT-2 | Static Pseudo User/Item Index | Item co-occurrence pattern
User preference pattern | ✘ | N/A |
| TIGER [35] | N/A | Static Semantic User/Item Index | Item co-occurrence pattern | ✘ | ✔ |
| LC-Rec [51] | LLaMA-2 | Static Semantic Item Index | Item co-occurrence pattern | ✘ | ✘ |
| ETEGRec [27] | T5 | Dynamic Semantic Item Index | Item co-occurrence pattern | ✔ | ✘ |
| STORE [29] | OPT-Base | Dynamic Semantic Item Index | Item co-occurrence pattern | ✘ | ✘ |
| TokenRec [32] | T5 | Static Semantic User/Item Index | Item co-occurrence pattern
User preference pattern | ✘ | ✘ |
| ED$^2$ (Ours) | LLaMA-2 | Dynamic Semantic User/Item Index | Item co-occurrence pattern
User preference pattern
User co-purchase pattern | ✔ | ✔ |

## D  Human Instruction Prompt

We summarize the human instruction prompts which are specially designed to utilize the high-order user-item interaction patterns.

---

**(i) User Co-Purchase Pattern**

**Prompt 1:** The item *<item>* has been historically clicked by users *<users>*. Can you predict another possible user which will click this item?.

**Prompt 2:** According to the users *<users>* that have clicked the item *<item>*, can you determine the next possible user wanting the same item?

**Prompt 3:** After clicked by these users *<users>*, who is the next user that may be keen on the item *<item>*?

**Prompt 4:** You have access to the item *<item>*'s historical user interaction record *<users>*. Now your task is to predict another possible user that loves the same item based on the past interaction.

**Prompt 5:** Considering the fact that several users *<users>* have clicked the same item *<item>*, forecast who is the next user that will be interested in this item.

---

**(ii) Implicit User Preference Pattern**

**Prompt 1:** Based on the user preference, what is the comment of user *<user>* on item *<item>*?

**Prompt 2:** Within the online shopping website, how does *<user>* feel after buying item *<item>*?

**Prompt 3:** I want to know how user *<user>* describes item *<item>*, could you inform me the description?

- - - - - - - - - - - - - - - - - - - - - - - - - - - - - -

**Prompt 4:** The user *<user>* searches for *<query>*, could you deduce what item he will like?

**Prompt 5:** Based on the user *<user>* current query *<query>*, please select the most suitable item for him.

**Prompt 6:** As a search engine, please answer the query *<query>* of user *<user>* by providing the possible item.

---

**(iii) Explicit User Preference Pattern**

**Prompt 1:** What is the preference of user *<user>*?

**Prompt 2:** As a recommender system, briefly summarize the preference of user *<user>*.

**Prompt 3:** How to describe the preference of user *<user>*?

---

## E  Dataset

We present the dataset statistics in Table 6. The three sequential recommendation datasets originate from the Amazon Product Review dataset [11], which contains user review data from May 1996 to October 2018. Particularly, three categories for the sequential recommendation task, i.e., *"Musical Instruments"*, *"Video Games"*, and *"Arts, Crafts and Sewing"*, are extracted and organized into individual dataset *Instruments*, *Games*, and *Arts*, respectively. Within the above datasets, each item is associated with a series of textual contents, including the item title, the detailed description, the item category, and so on. Similarly, the associated textual contents of the user entity include the user comment, the search query, and so on. Following standard procedure, inactive users/items with less then 5 interactions are filtered out and the user interactive sequence is created according to the chronological order.

**Table 6: Statistics of the evaluated datasets. *Avg.L* is the average length of the user interaction sequences.**

| Dataset | #User | #Item | #Interaction | Sparsity | Avg.L |
|---------|-------|-------|--------------|----------|-------|
| Instruments | 24,772 | 9,922 | 206,153 | 99.92% | 8.32 |
| Games | 50,546 | 16,859 | 452,989 | 99.95% | 8.96 |
| Arts | 45,141 | 20,956 | 390,832 | 99.96% | 8.66 |

## F  Baseline

Here we introduce the leading baseline recommendation models evaluated in the main experiment.

- **MLP-Based**: **FMLP-Rec** [53] proposes an all-MLP model with learnable filters for sequential recommendation, ensuring efficiency and reduces noise signals.
- **CNN-Based**: **Caser** [47] captures user behaviors by applying horizontal and vertical convolutional filters.

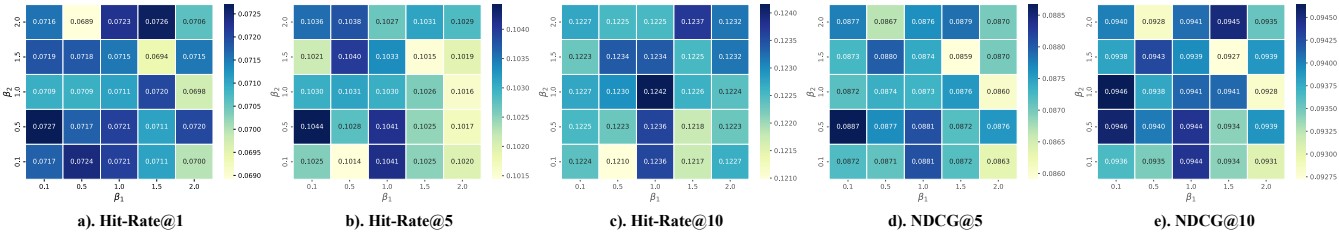

Figure 6: Parameter sensitivity analysis on the weighted hyper-parameters $\beta_1$ and $\beta_2$.

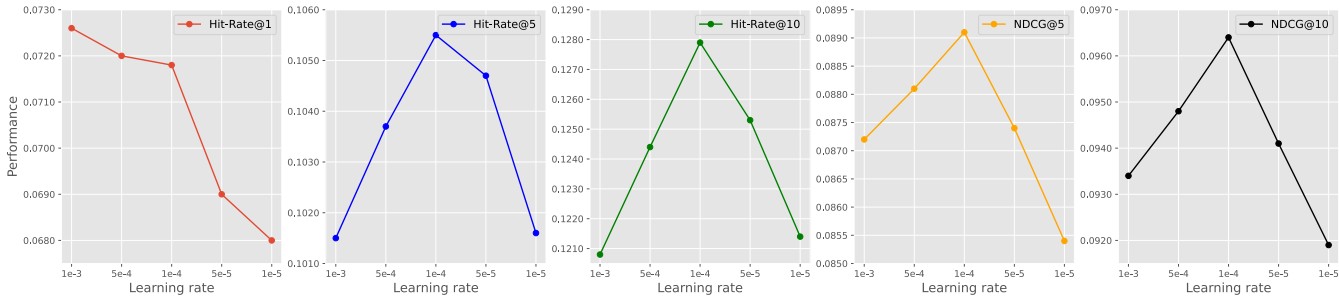

Figure 7: Parameter sensitivity analysis on the $ED^2$ recommender learning rate $\eta$.

- **RNN-Based**: **HGN** [30] utilizes hierarchical gating networks to capture both long-term and short-term user interests from historical behaviors. **GRU4Rec** [19] is an sequential recommendation model that utilizes GRU [3] to encode the item sequence.
- **GNN-Based**: **SR-GNN** [45] models session sequences as graph structured data, to capture complex transitions of items. **MA-GNN** [31] applies GNN to model the item contextual information within a short-term period and utilize a shared memory network to capture the long-range item dependencies.
- **Transformer-Based**: **BERT4Rec** [36] adopts a bidirectional Transformer model and combines it with a mask prediction task for the item sequences modeling. **SASRec** [22] exploits a unidirectional transformer model to capture the item sequences and predict the next item. **FDSA** [50] focuses on the transformation patterns between item features, modeling both item-level and feature-level sequences separately through self-attention networks. **S$^3$-Rec** [52] utilizes mutual information maximization to pre-train a self-supervised sequential recommendation model, learning the correlation between items and attributes.
- **PLM/LLM-Based**: **P5-CID** [8] organizes multiple recommendation tasks in a text-to-text format and models different tasks uniformly using the T5 [9] model. **TIGER** [35] adopts the generative retrieval paradigm for sequential recommendation and introduces a semantic ID to uniquely identify items. **CLLM4Rec** [54] proposes mixed prompting strategy based on heterogeneous tokens to fulfill sequential recommendation task. **LC-Rec** [51] extends the TIGER framework and further adopts LLaMA [41] as the recommender backbone.

## G Implementation Detail

All the experiments are finished on a machine with 8 *NVIDIA Tesla V100-SXM2-32GB* GPUs, 40 *Intel Xeon Platinum 8168 2.70GHz* CPUs, and Linux *Ubuntu 20.04.6* operating system.

We employ the open source large language model LLaMA [41] developed by Meta and introduce low-rank adaption technique

(LoRA) [16] for LLaMA efficient fine-tuning. The representation dimension of LLaMA model is 4096 and the original LLaMA vocabulary size is 32000. A language model head is appended to translating the hidden representation into the extended vocabulary. For the dual dynamic index generator, the dual index length $L$ is set as 4, and thus both the item and the user branch have 4 hierarchically cascade quantization codebooks, respectively. Each quantization codebook consists of 256 quantization embeddings whose dimension is set to 64. Therefore, the total number of the dual dynamic index tokens equals $2 \times 4 \times 256 = 2048$. The hierarchical variational quantization auto-encoder within the dual dynamic index generator contains a series of linear layers [4096, 2048, 1024, 512, 256, 128, 64], gradually projecting the LLaMA representation into the codebook hidden space and deducing the latent token distribution. We prepare a warm-up phase with a learning rate of 1e-3 for the dual dynamic index generator, by freezing the LLaMA-backbone recommender. For the $ED^2$ recommender training phase, we adopt the AdamW [23] optimizer with a learning rate of 5e-4.

## H Parameter Sensitivity Analysis

In this section, we conduct analysis towards the weighted parameters $\beta_1$ and $\beta_2$ of the index level alignment supervision and the token level alignment supervision, and the $ED^2$ learning rate $\eta$, .

### H.1 Weighted Hyper-Parameter

In Formula (11), $\beta_1$ and $\beta_2$ adjust the magnitude of the alignment supervision loss $\mathcal{L}_{\text{ID}}$ and $\mathcal{L}_{\text{Token}}$. We investigate the sensitivity of $ED^2$ recommender towards $\beta_1$ and $\beta_2$ and the evaluation result is presented in Figure 6. With $\beta_1$ and $\beta_2$ set as {0.1, 0.5, 1.0, 1.5, 2.0}, we notice that the best performance tends to be achieved when $\beta_1$ is less the $\beta_2$ (i.e., the lower triangle of the heatmap). This phenomenon implies that the token level alignment supervision (corresponding to $\beta_2$) is more difficult than the index level alignment supervision (corresponding to $\beta_1$). Furthermore, we note that the

standard deviation of the sequential recommendation performance is less than $9.7 \times 10^{-4}$, indicating that the $ED^2$ recommender is non-sensitive to the weighted parameter.

## H.2 Learning Rate

As shown in Figure 7, we evaluate the sequential recommendation performance of $ED^2$ recommender with the learning rate set as {1e-3, 5e-4, 1e-4, 5e-5, 1e-5}. The result reveals that the most suitable learning rate for $ED^2$ is around 1e-4. When the learning rate is larger than 5e-4 or less than 5e-5, the overall sequential recommendation performance depresses sharply. Different from other metrics, the Hit-Rate@1 keeps increasing with the learning rate growing. The phenomenon can be attributed to the difficulty of the Top-1 recommendation task which necessitates a large learning rate.

## I LSH Indexing Mechanism

To justify the superiority of the dynamic index mechanism, we introduce the dynamic locality sensitive hashing (LSH) index and the static LSH index as the compared variants. The locality sensitive hashing technique is design for the approximate nearest neighbor search, which is able to generate discrete index effortlessly. Specifically, we introduce $h$ random hyper-planes $\phi, \phi_2, \cdots, \phi_h$ to conduct random projection towards the input embedding $X$ (i.e., the textual representation provided by LLM). Based on the inner production result, we compute the LSH token of $X$ as follows,

$$\omega' = \sum_{i=0}^{h-1} 2^i \cdot \mathbb{I}\left(\phi_i^T X > 0\right), \tag{17}$$

with $\omega'$ ranging from 0 to $2^h - 1$. By repeating the operation of Formula (17) for $M$ times, we obtain the LSH index of $X$ as follows,

$$\Omega^{\text{LSH}} = <\omega_1', \omega_2', \cdots, \omega_M'> . \tag{18}$$

In our implementation, $h$ and $M$ are set as 8 and 4 respectively, to ensure the same expressive cardinality to the dual dynamic index.

## J Time Complexity

The $ED^2$ recommender mainly consists of the dual dynamic indexer and the LLM-backbone sequential recommender. Therefore, the time complexity of $ED^2$ includes the following two cascade portions.

### J.1 Dual Dynamic Indexer

The dual dynamic indexer contains two homogeneous branches for the index generation of users and items. Taking the user indexer as example, the shape of the input matrix $X$ is $\mathbb{R}^{B \times d}$, where $B$ is the input batch-size and $d$ is the LLM representation dimension. Within the variational quantizer, the input matrix $X$ is transformed into the statistic characteristics, i.e., the mean value $\mu$ and the variance $\Sigma$, of the dynamic index token distribution, defined as follows,

$$\mu = \text{MLP}_\mu(X), \Sigma = \text{MLP}_\Sigma(X). \tag{19}$$

The overall complexity of Formula (19) is $2 \times Bd$. The stochastic sampling operation according to the Guassian distribution $\mathcal{N}(\mu, \Sigma)$ consumes constant time in regard with $B$ and $d$. Therefore, the time complexity of the dual dynamic indexer is

$$\Theta_{\text{Indexer}} = 2Bd. \tag{20}$$

## J.2 LLM-Backbone Sequential Recommender

Given the batch-size $B$, the sequence length $L$, the shape of the input token sequence $S$ is $\mathbb{R}^{B \times L}$. The token embedding layer converts the token index to the continuous embedding $X$ via the matrix multiplication defined as follows,

$$X = \text{One\_Hot}(S) \cdot E, E \in \mathbb{R}^{V \times d}, \tag{21}$$

where $E$ is the token emebedding matrix, $V$ is the LLM vocabulary size, and $d$ is the LLM representation dimension. The time complexity of Formula (21) is $BLVd$. Taking the LLaMA model in our implementation as example, the embedding matrix $X$ is fed into the attention module defined as follows,

$$X = \text{Softmax}\left(\frac{QK^T}{\sqrt{d}}\right)V, X = XO, \tag{22}$$

$$Q = W_Q X, K = W_K X, V = W_V X, O = W_O X. \tag{23}$$

The time complexity of the LLaMA attention is

$$\Theta_{\text{Att}} = 4BLd^2 + 3B^2L^2 d. \tag{24}$$

Afterwards, the language model head projects the hidden state $X \in \mathbb{R}^{B \times L \times d}$ into the LLM vocabulary via a linear projection whose time complexity is $BLd$. Overall, the time complexity of the $ED^2$ recommender is

$$\Theta_{ED^2} = 2Bd + BLVd + 4BLd^2 + 3B^2L^2 d. \tag{25}$$

According to the Formula (25), the $ED^2$ time complexity is linear to the LLM vocabulary $V$ and is quadratic to the batch-size $B$, the sequence length $L$, and the LLM representation dimension $d$. Furthermore, we present a inference time comparison in Table 7.

**Table 7: Inference time comparison against the baselines.**

| Model | Backbone | Time (s) |
|---|---|---|
| SR-GNN | GNN | $6.21 \times 10^{-2}$ |
| MA-GNN | GNN | $3.45 \times 10^{-3}$ |
| CLLM4Rec | GPT-2 | $4.04 \times 10^{-3}$ |
| LC-Rec | LLaMA-2 | $5.93 \times 10^{-2}$ |
| $ED^2$ (Ours) | LLaMA-2 | $5.97 \times 10^{-2}$ |

## K Collision Handling Mechanism

It is worth noting that the hierarchical variational quantization formulated by Formulas (3)-(5) is unable to guarantee the index uniqueness, i.e., different items/users are allocated with distinct indices. Through the Gaussian sampling operation may result in index collision, the corresponding recommender system is still practicable with a fairly low collision rate [51]. In our practice, the collision rate is around $3 \times 10^{-4}$. However, on one hand, the collision rate is affected by the dataset scale, the model architecture, the optimization parameters, and so on, which is extremely uncontrollable. On the other hand, within some special recommender systems, the index uniqueness is significant and must be guaranteed [28, 35]. Inspired by the rehashing method in hash collision, we append an additional token into the dual dynamic index, representing the order inside the collision set [28, 35]. For example, two different items share the same dynamic index $<a1, b2, c3>$. Then, the dynamic indices will be remapped to $<a1, b2, c3, p_0>$ and $<a1, b2, c3, p_1>$, which scrupulously guarantee the index uniqueness.