# OpenReview forum: "Unleash LLMs Potential for Sequential Recommendation by Coordinating Dual Dynamic Index Mechanism"
_ACM.org/TheWebConf/2025/Conference — WWW 2025 Oral_

### Official Review · Reviewer_yoLm · 2024-11-05

**Novelty:** 6
**Technical Quality:** 6

**Review:**

This paper proposes a novel LLM sequential recommender system based on the dual dynamic index mechanism. The dual dynamic index based sequential recommender system assemblies the indexer and the sequential recommender into an unified streamline.  The indexer adopts a dual architecture which consists of two homogeneous discrete index generators, taking charge of the index generation of users and items. The proposed framework makes it practical for the LLMs to exploit the high-order user-item interaction patterns. Extensive experimental results on three public datasets demonstrate the effectiveness of the proposed approach.

Strength
1. Well-written and easy to understand.
2. Motivation is clear and makes sense.
3. The proposed dynamic indexing mechanism is novel, compared to previous static indexing method.
4. The proposed user-side indexing is novel, compared to previous methods that only focus on item-side indexing.

Weakness
1. The method is only applicable to llama-based sequential recommendation.
2. This method does indexing dynamically, so it needs to do indexing for every iteration/inference, limiting its efficiency compared with static methods.

**Questions:**

1. Can the proposed indexing method be applied to other recommendation settings, such as collaborative filtering?

**Reviewer Confidence:**

3: The reviewer is confident but not certain that the evaluation is correct

**Scope:**

4: The work is relevant to the Web and to the track, and is of broad interest to the community

---

### Official Review · Reviewer_mgTF · 2024-11-27

**Novelty:** 6
**Technical Quality:** 6

**Review:**

### Paper Summary
This work starts with the motivation to further bridge the gap between LLM and sequential recommendation, from the perspective of user/item id indexing strategy, on top of existing semantic id and traditional id approaches. It considered two factors: (1). existing
LLM-based sequential RSs mostly separate the index generation from the sequential recommendation and (2).  the neglect of the user-related information.

The paper then proposed the dual dynamic index to solve the factor (1), and the e2e pipeline with a newly proposed multi-grained token regulator (m-GTR) for factor (2). The experiments covered overall comparison table, ablation studies, and the case study.

### Pros
1. This work is a non-trivial step for investigating the gap between LLM and SR, from the indexing perspective. The user info is helpful, but only when the SR task is described in the instruction (correct me if i am wrong).
2. Several interesting observations from the experiments will be beneficial for future developments and research. For example, "Naive utilization of user-related information is profitless to sequential recommendation task." is also observed in another related work [1], and "specific instruction tuning for high-order user-item interaction patterns is crucial."
3. The writing is well-done and clear to me.

### Cons
1. I don't see any critical issue for this paper. But authors are encouraged to at least discuss these two most recent references. They both investigated a close and almost the same topic as this submission. Disclaimer: I have no conflict of interest on these two references.


[1]. Tan, Juntao, et al. "Towards llm-recsys alignment with textual id learning." arXiv preprint arXiv:2403.19021 (2024).
[2]. He, Zhankui, et al. "Reindex-Then-Adapt: Improving Large Language Models for Conversational Recommendation." arXiv preprint arXiv:2405.12119 (2024).

**Questions:**

See the Cons

**Reviewer Confidence:**

4: The reviewer is certain that the evaluation is correct and very familiar with the relevant literature

**Scope:**

4: The work is relevant to the Web and to the track, and is of broad interest to the community

---

### Official Review · Reviewer_FgHA · 2024-11-29

**Novelty:** 4
**Technical Quality:** 5

**Review:**

This paper proposes a new approach End-to-End Dual Dynamic (ED2) recommender for LLM-based sequential recommender systems. A dual dynamic index generator is proposed to absorb the semantic information within the textual feature and the collaborative information within the historical interaction. A multi-grained token regulator and a series of novel instruction tuning tasks are proposed to unleashing the potential of LLMs for exploiting the dynamic user/item index tokens and the high-order user-item interaction patterns. Extensive experiments on three public datasets demonstrate the superiority of ED2.

Strengths:
1.	The motivation is clear, deeply boosting the LLM comprehension capacity towards the dynamic index tokens is interesting and valuable.
2.	Each part of the method is aligned with the motivation. Part of the method has a solid theoretical foundation, such as the von Mises-Fisher distribution used in the proposed hierarchical variational quantization auto-encoder.
3.	The experiments and results are solid. ED2 achieves an average improvement of 19.62% in Hit-Rate and 21.11% in NDCG metric, which is a huge improvement.

Weakness:
1.	All the three datasets used in the Experiment section have particularly similar sparsity and average length. (Please refer to Questions 1)
2.	Subsection 2.4 of the Method section is not innovative and effective enough, some important implementation details are missing. (Please refer to Questions 2,3,4)

**Questions:**

1.	The datasets used by the authors have particularly similar sparsity and average length, however, longer length of the user interaction sequences seems to be helpful for “High-Order Interaction Pattern Exploitation”, in this case, the user co-purchase pattern will be more specific. Have the authors explored the effect of the length of user interaction sequences to ED2?
2.	In Subsection 2.4 of the Method section, the authors claimed that “The LLM-backbone recommender is prompted with several users historically interacting with the specific item, and then instructed to generate the index of another user who is fond of the same item”. However, there are no implementation details about how to select “several users”. Only use several users other than all users, will LLM fully understand the relations between users? What’s more, will the number of generated users have an impact on the results? The process of generating users for LLM seems to be very random, is further filtering required?
3.	In Subsection 3.3 of the Experiment section, the difference between w/o exploit and w/o user is very small on all metrics, as a reader, it's hard to believe that this part of the work is effective. The authors can conduct a case study, comparing the preferences of the high-order users predicted by LLM with the target users, which would help to convince the reader.
4.	For exploiting the high-order user-item interaction patterns, what is the SUPERIORITY and NECESSITY of using the method proposed in Subsection 2.4 compared to the GNN-based method without LLM?
5.	In Subsection 3.4.2 of the Experiment section, could the authors add a baseline’s experimental results for Case 1 and Case 3 to visually prove that ED2 is better than the baseline?

**Reviewer Confidence:**

4: The reviewer is certain that the evaluation is correct and very familiar with the relevant literature

**Scope:**

4: The work is relevant to the Web and to the track, and is of broad interest to the community

---

### Official Review · Reviewer_oCpA · 2024-11-30

**Novelty:** 4
**Technical Quality:** 5

**Review:**

Quality: The paper is technically sound, with comprehensive experiments and clear articulation of methods and results.

Clarity: The writing is well-structured, though some design choices (e.g., index length selection) could benefit from further explanation.

Originality: The dual dynamic index mechanism and multi-grained token regulator represent novel contributions to LLM-based recommendation.

Significance: The work addresses critical limitations in semantic-collaborative information integration, achieving state-of-the-art performance.

Pros：
- The proposed dynamic indexing tightly integrates semantic and collaborative information, addressing limitations of static indexing in existing LLM-based recommender systems.
- By introducing high-order interaction modeling through customized instruction tuning, the approach effectively captures nuanced user behaviors.
- The model demonstrates robust superiority over state-of-the-art methods across multiple datasets, achieving substantial gains in Hit-Rate and NDCG metrics.

cons:
- It seems that this article does not have released code. Without released code, reproducibility cannot be guaranteed.
- The paper proposes combining index generation with recommendation and dynamically updating the indices, which introduces higher computational overhead. Although the authors have implemented certain designs to reduce computational complexity, the paper still lacks a detailed description of the overall computational cost, leaving the reproducibility of this work uncertain.

**Questions:**

Is the length and hierarchical granularity of the dynamic index related to the specific requirements of the data scale or recommendation scenarios? During the design process, did you consider using shorter or longer index lengths, as well as different granularities of hierarchical quantization, and what were the trade-offs involved in these choices?

**Reviewer Confidence:**

2: The reviewer is willing to defend the evaluation, but it is likely that the reviewer did not understand parts of the paper

**Scope:**

4: The work is relevant to the Web and to the track, and is of broad interest to the community

---

### Official Review · Reviewer_rJTb · 2024-12-02

**Novelty:** 5
**Technical Quality:** 5

**Review:**

This paper focuses on the learnable semantic index for a sequential recommendation, which is achieved by the proposed End-to-end Dual Dynamic Index (ED$^2$) mechanism. Besides, index optimization, user information incorporation, and high-order information exploitation have also been considered by the proposed m-GTR, user-related instruction tuning tasks modules. Overall, this endeavor has contributed to the advancement of the learnable semantic index, yet the framework is excessively heavy, and the underlying motivations of the key contribution remain inadequately substantiated. The detailed reviews are as follows:

Strength:

1. The learning of end-to-end learnable semantic index is reasonable and meaningful, which is supposed to offer a harmonic integration of collaborative and semantic information.

2. Extensive experimental results have demonstrated the promising recommendation performance gain of ED$^2$, especially the significance of the learnable semantic index as verified in the ablation study.

Weakness:

1. Motivation. The author states their motivation for the dynamic index by the following example: *```For example, the film Transformers (July 3, 2007) and the teaching video Transformer Detailed Elaboration (October 28, 2021) are highly similar in terms of the textual contents, yet fractionally overlap within the user interaction records.```* However, in Section 3.4, the provided case studies can only prove that the index learned by ED$^2$ captures semantic information, which is trivial since the index is generated based on semantic information. A case study that proves ED$^2$ can avoid the *`Transformer & Transformer Detailed Elaboration`* issue is expected.

2. Technical Contribution. (1) The dynamic index generator is an RQ-VAE backbone[1] equipped with the von Mises-Fisher distribution[2]; (2) Codebook regulation through contrastive learning has been investigated in the literature[3]; (3) Pre-training tasks designed to explore user-related and higher-order information are similar to the main sequential recommendation task, thus appearing incremental and offering limited benefit to the research community.

3. Complexity. The framework is relatively heavy and complex, with many different components used. But in Section 3.3.1, we can find user-related (w/o user), and high-order (w/o exploit) information only brings marginal improvement. Besides, only theoretical computational complexity of training is provided. I'm concerned about the entire pre-training time, fine-tuning time, and GPU memory consumption of ED$^2$, especially considering the heavy pre-training tasks used to capture the user-related and high-order information.

4. Experiment Details. The adoption of a stronger backbone, llama-2, by ED$^2$ for capturing semantic information obscures the actual performance gains: whether they stem from the proposed strategy or simply the stronger backbone?

Typos:

1. <a1,b155,c222,**d138**> in Section 3.4.2 may be <a1,b155,c222,**d238**> as illustrated in Figure 5(a).

[1] Lee, Doyup, et al. "Autoregressive image generation using residual quantization." Proceedings of the IEEE/CVF Conference on Computer Vision and Pattern Recognition. 2022.

[2] Takida, Yuhta, et al. "Sq-vae: Variational bayes on discrete representation with self-annealed stochastic quantization." arXiv preprint arXiv:2205.07547 (2022).

[3] Wang, Wenjie, et al. "Learnable Item Tokenization for Generative Recommendation." Proceedings of the 33rd ACM International Conference on Information and Knowledge Management. 2024.

**Questions:**

Please refer to the weakness.

**Reviewer Confidence:**

3: The reviewer is confident but not certain that the evaluation is correct

**Scope:**

4: The work is relevant to the Web and to the track, and is of broad interest to the community